# Safety and efficacy of quick-soluble gelatin microparticles for transarterial embolization of the lower urinary tract: Preclinical study in a rabbit urinary bladder embolization model

Sunghoon Jeon[1,2], Keunho Kim[1], Cheolwon Choe[1], Juil Choi[1], Gun Lee[3], Chung-Do Lee[3], Hyeon-Jeong Moon[3], Jun-Gyu Park[3], Jin-kyung Kim[2], Namsoon Lee[1], Dongwoo Chang[1]*

**1** College of Veterinary Medicine, Chungbuk National University, Cheongju, Republic of Korea, **2** Haemaru Referral Animal Hospital, Seongnam, Republic of Korea, **3** Department of Veterinary Zoonotic Diseases, College of Veterinary Medicine, Chonnam National University, Gwangju, Republic of Korea

* dwchang@cbnu.ac.kr

**Data availability statement:** All relevant data are within the manuscript and its Supporting Information files.

## Abstract

Quick-soluble gelatin microparticles (QS-GMP) are emerging embolic agents under investigation for temporary vascular occlusion, offering reduced ischemic risk compared to permanent materials. The aim of this preclinical study was to evaluate the safety and efficacy of QS-GMP for transarterial embolization in a rabbit model of urinary bladder embolization. Twelve male New Zealand White rabbits underwent bilateral umbilical artery embolization using QS-GMP. Animals were assigned to four time-points (immediately, 3, 7, and 14 days post-embolization), with comprehensive assessments including clinical observations, hematologic and serum biochemical analysis, angiography, and histopathology. The procedure was technically feasible in all animals without intraoperative complications. Temporary hematuria and a transient decrease in body weight were observed post-procedure, both of which resolved spontaneously. Complete occlusion of the cranial vesical artery and absence of bladder wall perfusion were achieved immediately after embolization, followed by full recanalization at 3 days. Angiographic imaging at 7 and 14 days revealed transient hypervascularization of the bladder wall. Histopathological analysis showed marked edema, epithelial necrosis, and inflammatory infiltration at 3 and 7 days, with full urothelial regeneration observed at 14 days. No signs of ureteral or renal injury, or adverse systemic responses were detected. These findings suggest that QS-GMP may serve as a feasible option for temporary arterial occlusion in future veterinary lower urinary tract applications, although further long-term evaluation is warranted.

**Funding:** The author(s) received no specific funding for this work.

**Competing interests:** The authors have declared that no competing interests exist.

## Introduction

In humans, transarterial embolization (TAE) is a valuable treatment option for non-resectable tumors, particularly hepatocellular carcinoma [1]. It involves the deliberate endovascular occlusion of the tumor-feeders, thereby reducing tumor size and burden [2]. Originally introduced in 1974 as a treatment for refractory hemorrhages unresponsive to conventional therapies, its utility has since extended to oncological applications [3–5]. In veterinary medicine, TAE is used for effective hemorrhage control and as palliative treatment for unresectable tumors [6–10].

TAE has been used for the management of lower urinary tract malignancies in humans, especially for the alleviation of tumor-related intractable hematuria [3,11,12]. However, in human medicine, surgical resection remains the primary treatment option for lower urinary tract tumors; therefore, TAE is mainly used for intractable hematuria management rather than tumor control [3]. Conversely, in veterinary medicine, TAE has primarily been limited to prostatic malignancies [13]. A recent case report demonstrated the partial embolization of lower urinary tract tumor-feeders in veterinary patients, highlighting a reduction in tumor size and improvement in clinical signs [14]. Nevertheless, the use of TAE for managing bladder and urethral tumors remains limited in canines and felines owing to the risk of non-selective embolization, which may lead to bladder wall necrosis. Moreover, the small-diameter caudal vesical artery in these species often hinders superselective catheterization of the tumor-feeders, technically challenging targeted embolization.

Quick-soluble embolic agents have recently garnered attention as alternatives to permanent embolic agents in human medicine. These agents are designed for short-term vessel occlusion to minimize tissue necrosis, thereby reducing the risk of ischemia and non-target complications commonly associated with permanent embolic agents [15]. Imipenem/cilastatin has commonly been used as the embolic agent [16,17]. Biodegradable and bioabsorbable embolic agents, including quick-soluble gelatin microparticles (QS-GMP) and resorbable microspheres, have recently attracted developmental and applicational interest [18–21]. A study including 10 patients with acute gastrointestinal bleeding reported that hemostasis was successfully achieved in all patients without any major complications such as ischemic injury when using QS-GSP for embolization [18]. Similarly, a musculoskeletal embolization study including 71 patients reported no ischemic complications, except minor and transient events such as temporary skin discoloration [22]. These findings indicate that quick-soluble embolic agents provide embolization comparable to conventional agents and with minimal ischemic risk.

The introduction of quick-soluble embolic agents in veterinary practice could potentially facilitate palliative embolization in lower urinary tract tumors in veterinary patients. These embolic agents may enable tumor size reduction, minimize ischemic bladder and ureteral injury, and offer a safer and more effective therapeutic embolization. The potential notwithstanding, studies on the evaluation of quick-soluble embolic materials in veterinary medicine are lacking. Therefore, the aim of this preclinical study was to evaluate the efficacy and safety of embolization with QS-GMP in the lower urinary tract of a rabbit model of urinary bladder embolization. The findings

support its feasibility and short-term safety, providing a foundation for future clinical application in veterinary patients with lower urinary tract disorders.

## Materials and methods

### Animals

This study was designed as an exploratory pilot preclinical study. Therefore, a total of 12 animals (3 per time-point) was selected based on ethical feasibility and consistency with previous embolization feasibility study [20]. Twelve 5-month-old male New Zealand White rabbits (Hanabiotech, Pyeongtaek, Korea), weighing 3.3–4.1 kg, were used in this study. The rabbits were housed in individual cages and provided commercial dry food (Lab. Rabbit, Cargill Agri Purina, Pyeongtaek, Korea) and water ad libitum. The clinical condition of every rabbit was assessed via physical examination, complete blood count (CBC), serum biochemistry, urinalysis, and abdominal ultrasonography. The body weight of all rabbits was measured and recorded daily throughout the experimental period. Invasive procedures (blood collection, injection, and angiography) were performed under general anesthesia. All animal experimental procedures were approved by the Institutional Animal Care and Use Committee of Chungbuk National University (Protocol Number: CBNUA-24-0089-02), and complied with the ARRIVE 2.0 guidelines (https://arriveguidelines.org/arrive-guidelines).

### Bladder embolization procedure

All rabbits were administered 2 mg/kg tramadol hydrochloric acid (Tramadol, Dongkwang Pharm, Korea) and 20 mg/kg cefazolin sodium (Cefozol, Hankook Korus Pharm, Korea) intravenously. The rabbits were anesthetized with intramuscular administration of ketamine hydrochloride (15 mg/kg; Yuhan Ketamine 50 inj., Yuhan, Seoul, Korea) and medetomidine hydrochloride (0.25 mg/kg; Domitor, Zoetis Korea Ltd., Seoul, Korea). Anesthesia was maintained with isoflurane (Terrell, Piramal Critical Care, Bethlehem, USA) and oxygen via a face mask. A crystalloid fluid solution was administered intravenously (5 mL/kg/h) for intraoperative fluid management. Continuous monitoring included clinical evaluation, measurement of inspiratory and expiratory fractions of $CO_2$, $O_2$, and isoflurane, pulse oximetry, and indirect blood pressure measurement.

Post-anesthetization, the rabbits were placed in dorsal recumbency on a fluoroscopy table. Under sterile conditions, the bilateral carotid artery areas were clipped and prepared. A transarterial cut-down approach via the carotid artery was performed using a 4F introducer sheath (Prelude Ideal, Merit Medical, South Jordan, USA). A 4F-angled angiographic catheter (KMP, Jungsung Medical, Seoul, Korea) was advanced over a 0.035-inch guidewire (Zip Wire, Boston Scientific, Marlborough, USA) through the carotid artery and thoracic aorta up to the caudal aorta under mobile C-arm fluoroscopy (Veradius, Philips, Andover, USA) or ceiling-mounted angiography (Artis Q Ceiling, Siemens Healthineers, Erlangen, Germany) guidance (Fig 1). The guidewire was subsequently withdrawn, and under fluoroscopic guidance, 2 mL of contrast agent (Omnipaque 300°, GE Healthcare, Shanghai, China) was injected to visualize the vascular anatomy of the terminal aorta.

To access each common iliac trunk and umbilical artery, the bilateral umbilical arteries were superselectively catheterized using a 0.014-inch microwire (Meister S14, Asahi Intecc, Nagoya, Japan) and a 1.5F microcatheter (Veloute Ultra, Asahi Intecc, Nagoya, Japan). To minimize the risk of catheter-induced vasospasm, a 0.1 mL bolus of 1:10 diluted lidocaine hydrochloride (Huons Lidocaine HCl 20 mg/mL, Huons, Seongnam, Korea) and nitroglycerin (Nitrolingual 50 mg/mL, Unipharm Korea, Seoul, Korea) solution was infused immediately after microcatheter positioning in the umbilical artery. No arterial spasm was observed in any animal. For embolization of the umbilical artery, 100 mg of fish-derieved low-molecular gelatin microparticles (Smart-Gel Mirip 50–125 µm, PL Micromed, Yangsan, Korea) were dissolved in 5 mL of normal saline and mixed with 10 mL of contrast agent. The prepared embolic agent was aliquoted into 1 mL locking syringes. Under continuous fluoroscopic guidance, the embolic material was slowly infused into the target artery while carefully monitoring for any reflux using real-time conventional digital subtraction angiography

**Fig 1. Intra-operative conventional digital subtraction angiography (cDSA) (A, C–F) and cone-beam computed tomography (B) images illustrating the procedure of urinary bladder embolization in a rabbit model.** (A) Aortic angiogram depicting the right common (Rt. CIA), external (EIA), and internal (IIA) iliac arteries. (B) Three-dimensional volume rendering technique image: the umbilical artery (UA) originates from the ventromedial aspect of the CIA. (C) Roadmap image: advancing the microcatheter and microwire into the UA. (D) Umbilical angiogram revealing the cranial vesical artery (CVA) and deferential artery (DA) branches. The ventral branch of the DA (VB) courses caudally toward the testis. (E, F) Angiograms at the level of the CIA pre- (E) and post-embolization (F), respectively. The post-embolization image demonstrates an absence of blood flow through the UA to the urinary bladder.

(cDSA). Complete stasis was defined as the absence of antegrade contrast movement for at least 10 cardiac cycles despite heart pulsation. A post-embolization common iliac artery angiogram was performed to confirm target vessel occlusion. In cases where post-embolization cDSA demonstrated residual antegrade flow in the umbilical artery, additional embolic material was administered until the predefined stasis endpoint was achieved. Post-embolization, the microcatheter, angiographic catheter, and introducer sheath were removed, the carotid artery ligated, and the skin closed using standard surgical techniques. The rabbits were monitored daily, with regular assessments of body weight, appetite, water intake, and any abnormalities in defecation or urination. Assessments were performed at 4 designated time-points by euthanizing 3 animals at each interval: immediately post-embolization and 3, 7, and 14 days post-embolization (DPE). The allocation of animals to each time point was randomized to reduce selection bias. The rabbits euthanized immediately after embolization served as the control group for histopathological comparisons. Fig 2 illustrates the overall experimental schedule.

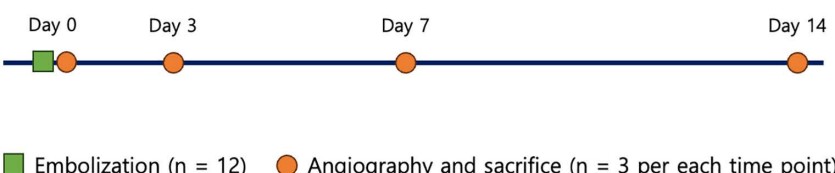

**Fig 2. Experiment schedule for embolization, angiography, and euthanization.**

## Hematological, serum biochemistry analysis and urinalysis

Ear vein blood samples were collected under sedation before anesthesia induction on the day of embolization (day 0). Hematological parameters included hematocrit (HCT, %) and white blood cell count (WBC, K/uL). Serum biochemistry parameters included blood urea nitrogen (BUN, mg/dL), creatinine (Cre, mg/dL), total protein (TP, g/dL), albumin (ALB, g/dL), aspartate aminotransferase (AST, U/L), and alanine aminotransferase (ALT, U/L).

At each designated time-point (3, 7, and 14 DPE), blood was collected immediately pre-euthanization to assess systemic responses and potential renal dysfunction post-embolization. HCT, WBC, BUN, and Cre were assessed for all animals at each experimental endpoint.

Samples were analyzed using an automated hematology analyzer (ProCyte Dx, IDEXX Laboratories Inc., Westbrook, USA) and a biochemical analyzer (Hitachi 3100, Hitachi High-Tech Corporation, Tokyo, Japan). Baseline values (day 0) were used as controls for subsequent comparative analyses.

Urine samples were aseptically collected via urethral catheterization on day 0 and at 3, 7, and 14 DPE. The catheter was gently inserted into the urethra under sedation to minimize trauma and contamination. A commercial urine dipstick test kit (UA Strips, IDEXX, Laboratories Inc., Westbrook, USA) was used, according to the manufacturer's protocol, to detect occult hematuria.

## Angiography

Angiography was performed pre-embolization, immediately post-embolization, and pre-euthanization. Selective umbilical artery angiography was performed bilaterally by injecting 0.7 mL of contrast agent via a 1.5F microcatheter. Angiographic cine images were acquired at 7.5 frames per second (pps) in the left anterior oblique lateral position (90°) using cDSA. The cDSA images were used to assess both cranial vesical artery patency and bladder wall microvasculature. The angiographic findings were classified using a 5-point scoring system and interpreted by a single veterinarian (Table 1 and S1 Fig). The scoring criteria for the cranial vesical artery and urinary bladder were adapted from similar preclinical embolization studies conducted in other organ models [20,23,24].

## Necropsy and histopathology

The rabbits were euthanized under anesthesia, either immediately post-embolization or post-angiography (3, 7, and 14 DPE), via intravenous administration of 5 mL of T-61 (T-61 injection, Intervet, Boxmeer, Netherlands). Post-euthanization, a midline laparotomy was performed for gross examination of the urinary bladder *in-situ*. Subsequently, the bladder was resected, and tissue specimens were collected for histopathological evaluation.

All samples were fixed in 10% neutral buffered formalin, processed routinely, and embedded in paraffin. Sections of 3–5 μm were prepared and stained with hematoxylin and eosin. A semi-quantitative histopathological scoring was applied to 6 distinct pathological features by a veterinary pathologist, with 15 years of experience, blinded to group allocation (Table 2 and S2 Fig). For each rabbit, 5 different regions of the tissue section were evaluated.

**Table 1. Angiographic findings scoring.**

| Score | Description |
|---|---|
| 0 | Complete absence of contrast blush in the bladder wall and cranial vesical artery |
| 1 | Moderate reduction in contrast blush in the bladder wall with cranial vesical artery patency |
| 2 | Normal contrast blush in the bladder wall with cranial vesical artery patency |
| 3 | Mild hypervascularization in the bladder wall with increased neovascularization |
| 4 | Dense contrast blush with marked hypervascularization in the bladder wall |

Each cranial vesical artery was independently assessed by a veterinarian, yielding 6 measurements per time-point.

**Table 2. Histopathologic findings scoring.**

| Category | Score | | | | |
|---|---|---|---|---|---|
| | 0 | 1 | 2 | 3 | 4 |
| Congestion and edema | 0 | Mild vascular dilation with minimal congestion and no notable edema | Moderate congestion with slight edema | Marked vascular engorgement with visible interstitial edema and tissue separation | Severe and diffuse congestion with extensive edema and widespread disruption of tissue architecture |
| Inflammatory cell infiltration | 0 | Rare, 1–5/phf | 5–10/phf | Heavy infiltrate | Packed |
| Necrosis | 0 | Minimal | Mild | Moderate | Severe |
| Fibroblast proliferation/ Fibrosis | 0 | Narrow band | Moderately thick band | Thick band | Extensive band |
| Neovascularization | 0 | Minimal capillary proliferation, focal, 1–3 buds | Groups of 4–7 capillaries with supporting fibroblastic structure | Broad band of capillaries with supporting fibroblastic structures | Extensive band of capillaries with supporting fibroblastic structures |
| Epithelial regeneration | 0 | Minimal regeneration with scattered immature epithelial cells | Partial epithelial coverage with disorganized structure | Near-complete regeneration with mild atypia | Fully regenerated epithelium with normal histological architecture |

For each rabbit, 5 different regions of the tissue section were independently evaluated by a veterinary pathologist.

phf: per high-powered (400x) field.

## Statistical analysis

All statistical analyses were performed using a spreadsheet (Microsoft Excel 2019, Microsoft, Redmond, USA) and statistical software (SPSS Version 29.0.2.0, IBM Corp., Armonk, USA). Body weight, serum biochemical parameters, angiographic score, and histopathologic score are expressed as mean ± standard deviation, and the volume of embolic slurry is expressed as medians and ranges. The median and interquartile range (IQR) were calculated for angiographic and histopathologic scores. The Kruskal-Wallis test was used to compare the serum biochemical parameters on the euthanization days (immediately post-embolization and 3, 7, and 14 DPE). Angiographic and histopathologic scores were compared between day 0 and each time-point (3, 7, and 14 DPE) using the Mann–Whitney U test. Statistical significance was set at $p < 0.05$.

## Results

### Animals

All animals underwent a single embolization procedure on day 0. A decrease in mean body weight was observed at 3 and 7 DPE, which increased at 14 DPE (S3 Table 1). The rabbits began to gain weight between 8 and 10 DPE.

Gross hematuria was observed in most animals between day 0 and 3 DPE. One animal exhibited hematuria at 13 DPE (S4 Table 2). No other clinical signs, such as lethargy or pain-related responses, including slow gait, decreased activity, or ear drooping, were observed during the study period.

## Bladder embolization procedure

Bilateral umbilical artery embolization was successfully completed without any procedural complications, and complete vascular stasis was achieved in all animals. In every case, the microcatheter could be advanced to the umbilical artery level bilaterally, and embolic material was infused consistently at this location. The median volume of the embolic slurry was 0.2 mL (range: 0.1–0.4 mL) and 0.3 mL (range: 0.2–0.4 mL) on the right and left sides, respectively, with a total median volume of 0.525 mL (range: 0.3–0.8 mL).

## Hematological, serum biochemistry, and urine analysis

Prior to embolization, all rabbits exhibited no clinical signs, and CBC and serum biochemical parameters were within the normal reference ranges (Table 3). Post-embolization, the euthanization time-point comparisons between day 0 and 3, 7, and 14 DPE revealed fluctuations in HCT, WBC, and Cre values. Although WBC and BUN differed significantly across groups in the Kruskal–Wallis test (WBC: $p = 0.043$; BUN: $p = 0.023$), WBC values consistently remained within their physiological reference ranges. The mean BUN levels transiently exceeded the upper reference limit at 3 and 7 DPE; however, they returned to within normal limits by 14 DPE.

Occult hematuria was consistently observed in all rabbits on 3 and 7 DPE. On 14 DPE, occult hematuria was absent in 1 rabbit, whereas the remaining animals continued to exhibit positive results, despite the presence of gross hematuria.

## Angiography

Immediate post-embolization cDSA images revealed complete occlusion of the cranial vesical artery and absent bladder wall blush in all rabbits. During follow-up angiography, bilateral umbilical arteries were consistently visualized at all time-points, and microcatheter advancement to the umbilical artery level was feasible in every animal without technical difficulty. At 3 DPE, cDSA images revealed complete recanalization of the cranial vesical artery and reduced microvascularization of the bladder wall (Fig 3). At 7 DPE, cDSA images revealed distinct hypervascularization of the bladder wall and an enlarged diameter of the cranial vesical artery. At 14 DPE, hypervascularization was mildly reduced compared with that on 7 DPE; however, increased neovascularization compared with that in the pre-embolization images was observed. The patency of the cranial vesical artery at 3, 7, and 14 DPE was confirmed using the cDSA images. The angiographic scores are summarized in Table 4. The angiographic scores of each time-point revealed statistically significant differences compared with those pre-embolization.

**Table 3. Hematological and serum biochemical parameters of the rabbits on day 0 and euthanization days.**

| Parameter (ref. range) | Day 0 (n = 12) | Day 3 (n = 3) | Day 7 (n = 3) | Day 14 (n = 3) |
|---|---|---|---|---|
| HCT (31–50%) | 38.34 ± 3.08 | 35.87 ± 3.45 | 37.97 ± 7.09 | 34.73 ± 7.91 |
| WBC (5.9–18.3 K/μL)* | 9.3 ± 2.0 | 16.98 ± 8.12 | 12.13 ± 1.48 | 11.10 ± 2.00 |
| BUN (10–30 mg/dL)* | 22.4 ± 5.92 | 36.10 ± 5.37 | 31.17 ± 9.27 | 19.63 ± 2.31 |
| Cre (0.5–1.5 mg/dL) | 0.98 ± 0.31 | 1.32 ± 0.11 | 1.13 ± 0.32 | 1.13 ± 0.33 |
| TP (5.4–7.5 g/dL) | 6.72 ± 0.62 | ND | ND | ND |
| ALB (2.5–4.0 g/dL) | 2.05 ± 0.12 | ND | ND | ND |
| AST (15–45 U/L) | 20.25 ± 10.76 | ND | ND | ND |
| ALT (10–55 U/L) | 33.42 ± 10.04 | ND | ND | ND |

Data are presented as mean ± standard deviation.

Kruskal-Wallis test (HCT, WBC, BUN, Crea), *p < 0.05.

HCT, hematocrit; WBC, white blood cell count; BUN, blood urea nitrogen; Cre, creatinine; TP, total protein; ALB, albumin; AST, aspartate aminotransferase; ALT, alanine aminotransferase, ND, not done.

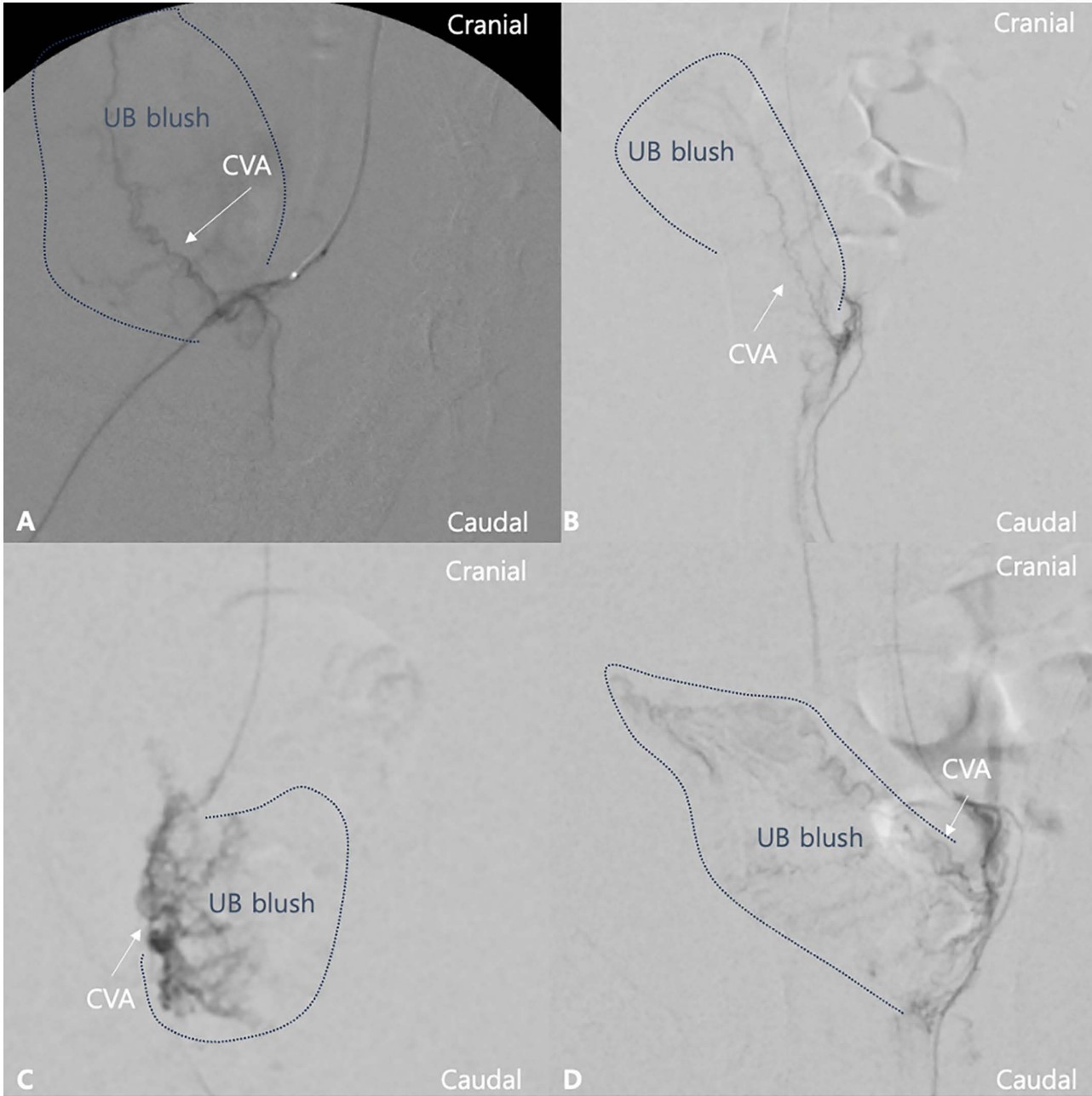

**Fig 3. Representative angiographic images at day 0 (A, pre-embolization), 3 days post-embolization (DPE) (B), 7 DPE (C), and 14 DPE (D).** (A) The cranial vesical artery (CVA) and normal microvasculature supplying the bladder wall are clearly visualized with normal urinary bladder (UB) blush (blue dotted line). (B) 3 DPE: recanalization of the cranial vesical artery is observed; however, the microvascularization of the bladder wall is reduced compared with that on day 0. (C) 7 DPE: the bladder lumen appears collapsed, and marked enlargement of the cranial vesical artery with distinct hypervascularization of the bladder wall is visible. (D) 14 DPE: the degree of hypervascularization is mildly reduced compared with 7 DPE.

**Table 4. Angiographic scoring analysis.**

| | Day 0 (n=6) (PRE) | Day 0 (n=6) (POST) | Day 3 (n=6) | Day 7 (n=6) | Day 14 (n=6) |
|---|---|---|---|---|---|
| Mean±SD | 2.00±0.00 | 0.00±0.00 | 1.33±0.52* | 3.83±0.41** | 2.67±0.52* |
| Median (IQR) | 2.00 (0.00) | 0.00 (0.00) | 1.00 (0.75) | 4.00 (0.00) | 3.00 (0.75) |

Each group comprises 6 vessels (bilateral umbilical arteries in 3 animals per group).

Score 0, complete absence of contrast blush in the bladder wall and cranial vesical artery; 1, moderate reduction in contrast blush in the bladder wall with cranial vesical artery patency; 2, normal contrast blush in the bladder wall with cranial vesical artery patency; 3, mild hypervascularization in the bladder wall with increased neovascularization; 4, dense contrast blush with marked hypervascularization in the bladder wall.

PRE, pre-embolization; POST, post-embolization; SD, standard deviation; IQR, interquartile range

Mann–Whiteney U test. *$p<0.05$ and **$p<0.01$ compared with day 0 (PRE).

## Necropsy and histopathology

At necropsy, no gross abnormalities were observed in the urinary bladder wall or cranial vesical arteries at day 0 (Fig 4). At 3 DPE, the cranial vesical arteries appeared congested, with moderate to severe erythema of the urinary bladder wall. At 7 DPE, the cranial vesical arteries appeared distinctly enlarged and tortuous, and the bladder wall was markedly thickened and collapsed. Focal yellow necrotic lesions were observed in 2 rabbits. Ascites or bladder wall perforations were not noted in any of the rabbits. At 14 DPE, the cranial vesical vessels remained dilated; however, the bladder wall exhibited recovery with mild erythema, and the bladder lumen was distended. No morphological abnormalities of the testes, ductus deferens, or kidneys, nor any evidence of ureteral injury, dilatation, or hydronephrosis were observed.

Histopathology revealed severe congestion in the mucosal and submucosal layers of the urinary bladder, and neither epithelial dysplasia nor exfoliation was detected in the mucosal lining on day 0 (Fig 5). The injected embolic agents were found within the vessels. At 3 DPE, severe edema was observed in the mucosal and submucosal layers, with dense infiltration of lymphoid cells, including neutrophils. The transitional epithelium exhibited coagulative necrosis followed by desquamation, indicative of acute injury. At 7 DPE, persistent inflammation, accompanied by active neovascularization and fibroblast proliferation, was observed, suggesting progression to chronic inflammation. Additionally, the regenerative change of transitional epithelium was noted. At 14 DPE, bladder wall edema was reduced to mild levels, and the presence of hyaline-like basophilic deposits, sustained neovascularization. The transitional epithelium had completely recovered, demonstrating restoration of mucosal integrity. The histopathologic scoring results are summarized in Table 5.

## Discussion

This study established a rabbit umbilical artery embolization model and techniques to assess embolic efficacy and safety of temporary vessel occlusion when using QS-GMP. Although various animal models have been developed to study therapeutic embolization, including models for aneurysms, cerebral ischemia, hepatic tumors, prostatic hyperplasia, the uterus, and the kidneys, a pre-clinical model specifically targeting embolization for the urinary bladder remains unestablished to date [25–33]. Previous embolization models, such as those targeting cerebral ischemia, hepatic, or renal arteries, have been primarily designed to study ischemia–reperfusion injury or to evaluate chemoembolization strategies. In the urogenital system, prostatic artery embolization models have been proposed for benign prostatic hyperplasia, however, they are limited to prostate-targeted occlusion and do not adequately represent bladder-directed vascular procedure [33]. Therefore, the present model fills this gap by offering a dedicated platform for urinary bladder–specific embolization research. Both angiographic and histopathologic evaluations confirmed effective bladder wall arterial occlusion. The procedure was technically feasible without any procedural complications. While this study was primarily designed for future veterinary clinical application the reproducibility and standardized protocol indicate that it may also serve as a useful preclinical animal model for embolization research.

In the present study, we introduced an experimental rabbit model and interventional approach for bladder embolization. Rabbit models enable easy to arterial access via carotid, femoral, and auricular artery, and selective embolization of

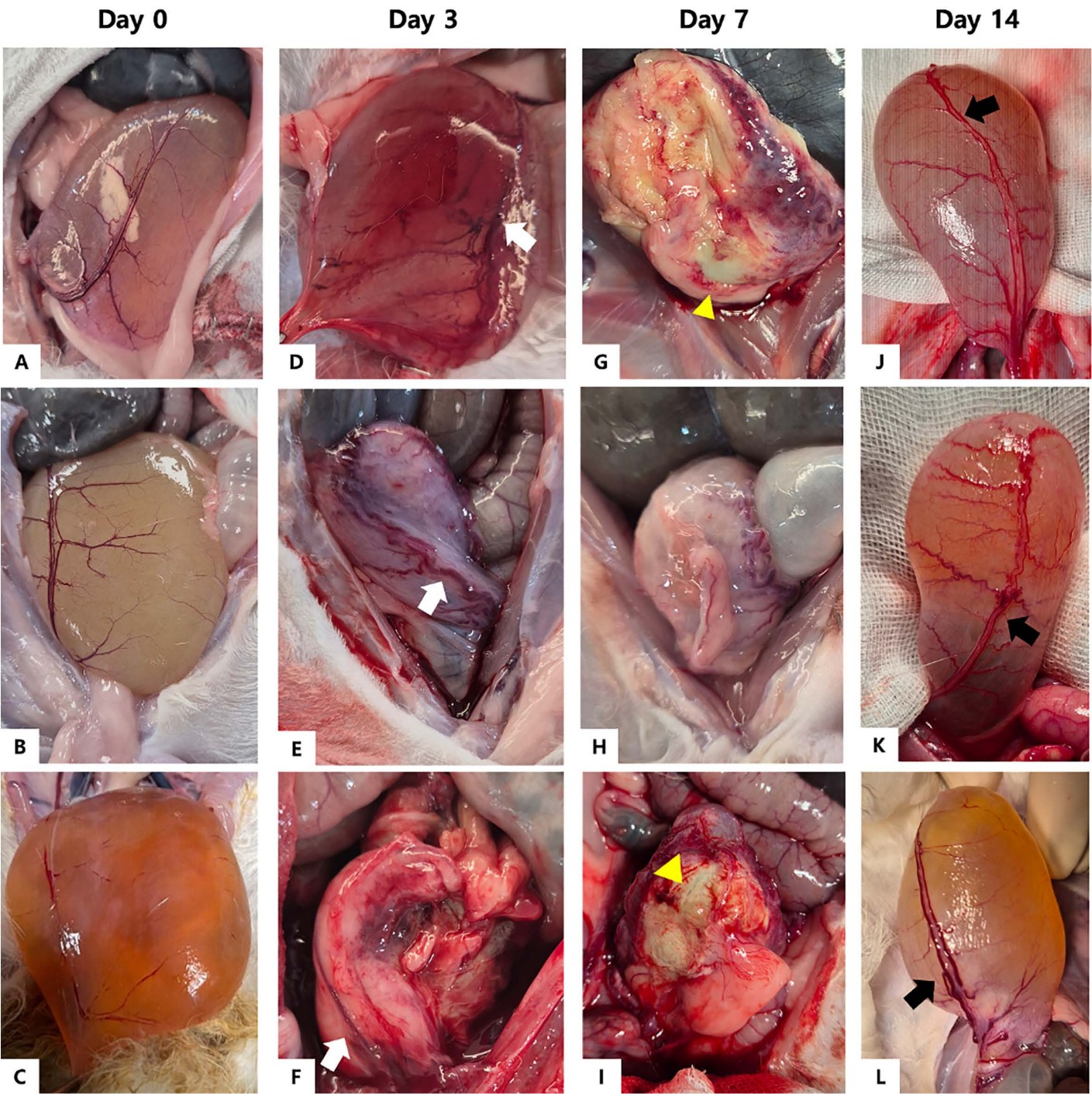

**Fig 4. *In-situ* images of the urinary bladder at necropsy from all rabbits at each time-point.** (A – C) Day 0: No gross abnormal lesion in the urinary bladder. (D -F) DPE 3: Cranial vesical artery and vein are dilated (white arrows) with reddish moderate to severe erythema of the urinary bladder wall. (G – I) DPE 7: Cranial vesical vessels are distinctly enlarged, and the bladder wall is thickened and collapsed. (G and I, yellow arrow heads) Focal yellow necrotic lesions. (J – L) DPE 14: Cranial vesical vessels remain dilated (black arrows); however, the bladder wall exhibits recovery with mild erythema.

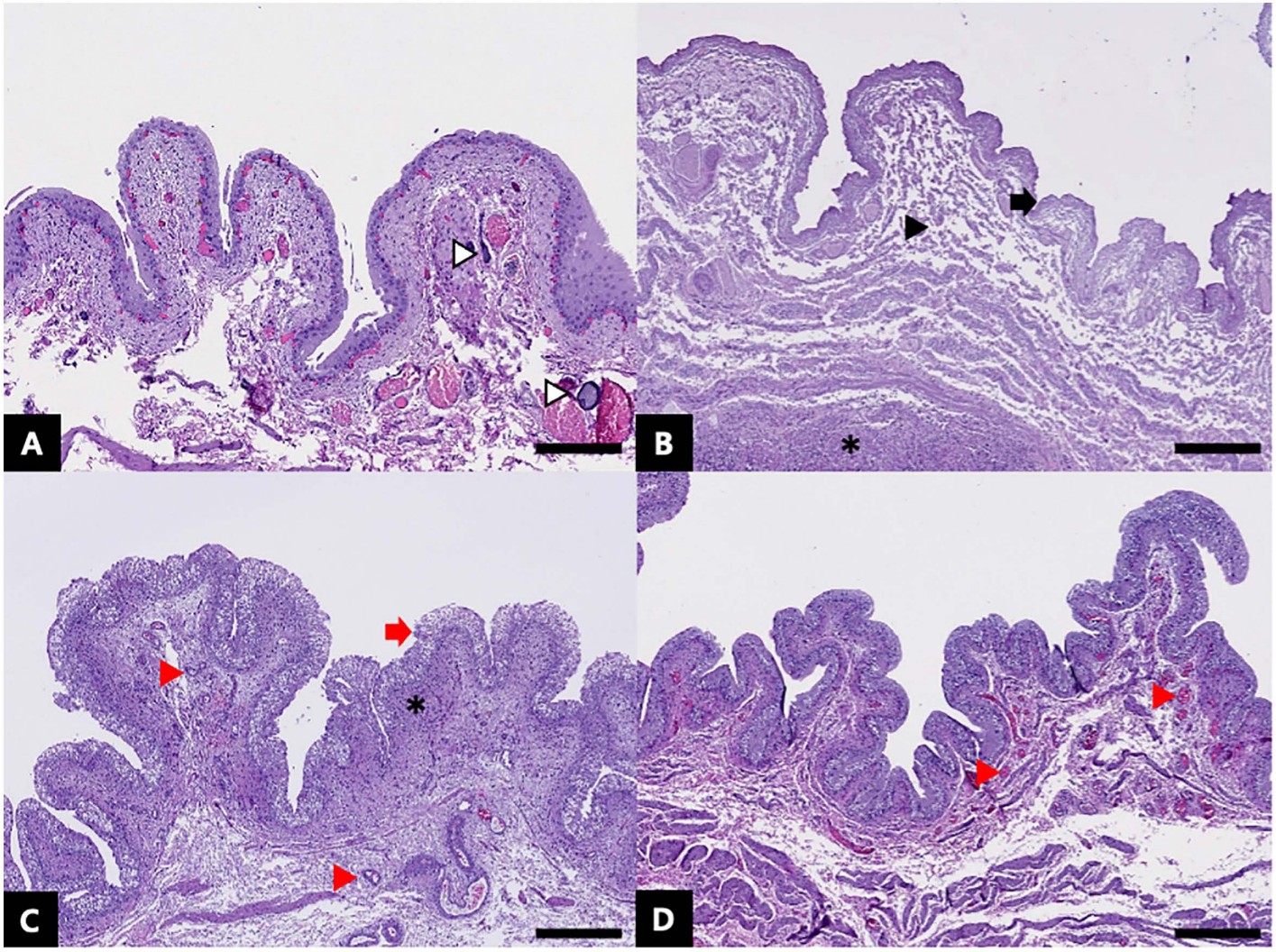

**Fig 5. Representative images of tissue sections at various time-points post-embolization.** (A) Tissue collected immediately post-embolization (day 0). Injected embolic materials (blank arrowheads) are noted. Bar = 200 μm (B) Tissue collected 3 DPE. Transitional cell degeneration and desquamation (arrow), necrosis of submucosa (arrowhead), and lymphoid cell infiltration (asterisk) are noted. Bar = 400 μm (C) Tissue collected 7 DPE. Regenerated transitional cells (red arrow), lymphoid cell infiltration (asterisk) and neovascularization (red arrowheads) are noted. Bar = 400 μm (D) Tissue collected 14 DPE. Transitional cells are fully regenerated with neovascularization (red arrowheads). Bar = 300 μm. These samples illustrate the temporal progression of histopathological changes post-embolization.

the umbilical artery, which is impossible in small animals such as rat or mouse. Additionally, rabbits are relatively easy to handle and maintain, and they pose fewer ethical and regulatory constraints compared with larger species such as dogs or pigs.

In this study's animal model, embolization was performed at the level of the umbilical artery rather than the cranial vesical artery. This decision was based on pre-experiment in-house pilot studies, which revealed that a 1.5F microcatheter could not be advanced into the cranial vesical artery. The microcatheter either became wedged or failed to select the cranial vesical artery, and in some cases, vascular perforation occurred during the injection of contrast agent or embolic agents. In male rabbits, the cranial vesical artery, a branch of the umbilical artery, is the primary blood supplier to the

**Table 5. Semi-quantitative scoring analysis of histopathologic findings.**

| Category | | Day 0 (n = 15) | Day 3 (n = 15) | Day 7 (n = 15) | Day 14 (n = 15) |
|---|---|---|---|---|---|
| Congestion and edema | Mean ± SD | 3.27 ± 0.7 | 3.27 ± 0.46 | 0.73 ± 0.59* | 1.73 ± 0.8* |
| | Median (IQR) | 3 (1.0) | 3 (0.5) | 1 (1.0) | 2 (1.0) |
| Inflammatory cell infiltration | Mean ± SD | 0.00 ± 0.00 | 3.73 ± 0.59* | 3.0 ± 0.65* | 0.8 ± 0.77* |
| | Median (IQR) | 0 (0.0) | 4 (0.0) | 3 (0.0) | 1 (1.0) |
| Necrosis | Mean ± SD | 0.00 ± 0.00 | 3.67 ± 0.62* | 2.4 ± 0.74* | 0.73 ± 0.70* |
| | Median (IQR) | 0 (0.0) | 4 (0.5) | 2 (1.0) | 1 (1.0) |
| Fibroblast proliferation/ Fibrosis | Mean ± SD | 0.00 ± 0.00 | 2.73 ± 0.80* | 2.8 ± 0.56* | 2.0 ± 0.65* |
| | Median (IQR) | 0 (0.0) | 3 (1.0) | 3 (0.5) | 2 (0.0) |
| Neovascularization | Mean ± SD | 0.00 ± 0.00 | 0.00 ± 0.00 | 2.93 ± 0.59* | 3.8 ± 0.41* |
| | Median (IQR) | 0 (0.0) | 0 (0.0) | 3 (0.0) | 4 (0.0) |
| Epithelial regeneration | Mean ± SD | 0.00 ± 0.00 | 0.00 ± 0.00 | 2.0 ± 0.53* | 3.73 ± 0.46* |
| | Median (IQR) | 0 (0.0) | 0 (0.0) | 2 (0.0) | 4 (0.5) |

Data represent mean ± SD for each group (n = 15 data points per time-point, derived from 3 rabbits, and 5 regions per rabbit).

SD, standard deviation; IQR, interquartile range.

Mann–Whitney U test, *$p < 0.001$ compared with day 0.

urinary bladder [34]. The umbilical artery usually emerged from the common iliac artery. The deferential artery, which supplies the ductus deferens, originates from the umbilical artery as well. In contrast, in female rabbits, the uterine artery originates from the umbilical artery, along with the cranial vesical artery and vaginal branches [35]. While the vagina may receive collateral perfusion from branches of the internal iliac artery, the uterus is predominantly supplied by the uterine artery, and embolization at the level of the umbilical artery could result in unintended uterine ischemia or necrosis. To avoid this confounding risk and to ensure vascular consistency during embolization, only male rabbits were selected for this study. Considering these technical limitations, the embolic agent was administered via the umbilical artery, which resulted in embolization of the deferential artery as well in male rabbits. However, gross necropsy findings in all rabbits revealed no evidence of ischemic damage to the ductus deferens or testes. The ductus deferens is primarily supplied by the testicular artery from the aorta and possesses its own vascular branch, providing collateral circulation, which may have contributed to perfusion preservation in this region [36–38].

Although a slight decrease in body weight was observed in all rabbits up to 7 DPE, the body weight began to increase beyond 8 DPE. The initial weight loss may be attributable to post-procedural discomfort and decreased food intake. Typically, embolization induces a post-embolization syndrome characterized by inflammation and tissue necrosis, with common symptoms including pain and vomiting, which may have contributed to the observed weight loss [39,40]. Additionally, an increase in WBC was observed post-procedure, which may support the presence of post-embolization systemic inflammatory responses. However, the recovery in body weight and WBC levels at 14 DPE indicates the transience and tolerability of these post-embolization effects. Regarding serum biochemistry profiles, BUN levels significantly increased post-embolization at 7 and 14 DPE. However, necropsy revealed no morphological abnormalities in either kidney, and no evidence of ureteral injury, dilation, or hydronephrosis was noted. Hence, these changes were considered transient.

Rapidly biodegradable embolic agents include Imipenem/cilastatin, QS-GMP, and resorbable microspheres [21]. Unlike conventional temporary embolic agents such as high-molecular-weight gelatin sponge particles, which typically degrade in 2–6 weeks, these agents dissolve within 30 min to 2 days, allowing early recanalization [16]. The QS-GMP used in the present study comprised low-molecular-weight, fish-derived gelatin. The low-molecular-weight gelatin particles dissolve at lower temperatures than conventional high-molecular-weight gelatin particles, which contributes to their enhanced biodegradability [20]. According to the manufacturer's datasheet, QS-GMP maintains embolic occlusion for approximately

12–24 h. This is supported by an in vitro study demonstrating that, under physiological temperature conditions (37 °C in saline), peripheral transparency of the particles began within 5 minutes on microscopy, indicating early surface dissolution [20]. In comparison with conventional high-molecular-weight gelatin sponge particles, QS-GMP exhibited markedly faster disintegration. Angiographic and histopathologic findings in the present study indicated complete vascular recanalization by 3 DPE. Epithelial tissue damage and bladder wall inflammation observed at 3 and 7 DPE were considered part of the inflammatory response associated with ischemic injury and tissue regeneration. Moreover, at 14 DPE, the urothelium had completely regenerated, indicating full structural recovery of the affected tissue.

Angiographic findings of increased microvascularization of the bladder wall at 7 and 14 DPE, compared with those at pre-embolization, were consistent with the observed histopathologic findings. A previous human study evaluating serial dynamic CTs after hepatic TAE reported a transient increase in arterial perfusion between days 2 and 6 post-procedure, which was attributed to acute inflammatory responses post-embolization, including vasodilation and increased vascular permeability [41]. Although the physiological environment of the bladder differs from that of the liver, particularly in terms of blood supply and local immune response, the transient vascular enhancement observed in this study may represent either reactive angiogenesis following temporary ischemia or an inflammatory hyperemic response. Further investigation, such as perfusion imaging or molecular assessment of angiogenic markers, would be required to distinguish between these mechanisms.

In the present study, procedural complications were limited to transient weight loss and hematuria. No major complications such as bladder perforation, uroabdomen, or ureteral obstruction were observed. These favorable safety outcomes may be attributed to the quick-soluble embolic agent. The QS-GMP used began to dissolve approximately 12 h post-procedure and enabling recanalization within 48 h. Although initial ischemic damage was observed in the urinary bladder wall, the restoration of perfusion through the parent artery possibly facilitated tissue recovery. Histopathologic evaluation revealed edematous change and inflammatory cell infiltration in the bladder epithelium at 3 and 7 DPE, with some necrotic changes noted at 7 DPE; however, complete epithelial regeneration was confirmed at 14 DPE. Human clinical research on embolization for acute gastrointestinal bleeding demonstrated that ischemic complications occurred in approximately 14.8% of patients treated with conventional GSP, whereas no such complications were observed when QS-GSP was used [18,42]. Similarly, a study on human genicular artery embolization revealed that transient skin color changes occurred in 51% of procedures, possibly due to early arterial recanalization following degradation of the quick-soluble embolic agent [22].

This study had several limitations. First, statistical analyses involving dependent variables were not feasible owing to the small sample size, which limited subgroup comparisons. Second, serial angiographic evaluations in a single rabbit were not possible. To evaluate both angiographic and histopathologic changes at defined end-points, the data analyses were conducted at different time-points. Even if serial angiography had been performed in the same rabbit, the corresponding histological data could not have been obtained simultaneously. Therefore, the animals were divided by time-point, allowing for endpoint-specific evaluation rather than longitudinal assessment in a single rabbit. Third, this study lacked a reference group using permanent embolic agents, which would have provided a valuable comparison for evaluating the efficacy and safety of the QS-GMP used in this study.

In conclusion, QS-GMP embolization of the umbilical artery was feasible and well-tolerated, supporting its potential use in veterinary lower urinary tract interventions and as a standardized preclinical embolization model for investigating therapeutic embolization strategies.

## Supporting information

**S1 Fig. Representative angiographic images illustrating the scoring criteria used for quantitative evaluation.** (A) Score 1: Moderate reduction in contrast blush in the bladder wall with cranial vesical artery patency. (B) Score 2: Normal contrast blush in the bladder wall with cranial vesical artery patency. (C) Score 3: Mild hypervascularization in the bladder wall with increased neovascularization. (D) Score 4: Dense contrast blush with marked hypervascularization in the bladder wall. (TIFF)

**S2 Fig. Representative histologic images illustrating the scoring criteria.** (A) Congestion and edema: Severe congestion observed in the mucosal and submucosal layers of the urinary bladder. (B) Inflammatory cell infiltration: The lamina propria and submucosa show evident vascular structural damage, accompanied by perivascular infiltration of inflammatory cells. (C) Necrosis: Severe necrosis around the vascular damage region. (D) Fibroblast proliferation: Severe infiltration of lymphoid cells was present in the mucosa, submucosa, and muscular layers, accompanied by the deposition of basophilic hyaline-like materials, neovascularization, and fibroblast infiltration (E) Neovascularization: Both neovascularization and the formation of mature blood vessels are evident. (F) Epithelial regeneration: The transitional epithelium of the bladder showed complete recovery.
(TIFF)

**S3 Table. Body weight of the rabbits on each euthanization day.**
(DOCX)

**S4 Table. Rabbits with gross and occult hematuria.**
(DOCX)

## Author contributions

**Conceptualization:** Sunghoon Jeon, Dongwoo Chang.

**Data curation:** Sunghoon Jeon, Keunho Kim, Cheolwon Choe, Juil Choi, Gun Lee, Chung-Do Lee, Hyeon-Jeong Moon, Jun-Gyu Park, Jin-kyung Kim, Namsoon Lee.

**Formal analysis:** Sunghoon Jeon, Keunho Kim.

**Investigation:** Sunghoon Jeon, Cheolwon Choe, Juil Choi, Chung-Do Lee, Hyeon-Jeong Moon, Jun-Gyu Park, Jin-kyung Kim, Namsoon Lee, Dongwoo Chang.

**Methodology:** Sunghoon Jeon, Gun Lee, Dongwoo Chang.

**Validation:** Sunghoon Jeon, Keunho Kim, Cheolwon Choe, Juil Choi, Gun Lee, Chung-Do Lee, Hyeon-Jeong Moon, Jin-kyung Kim, Dongwoo Chang.

**Writing – original draft:** Sunghoon Jeon.

**Writing – review & editing:** Jun-Gyu Park, Namsoon Lee, Dongwoo Chang.

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
