## [Decision Letter · Decision Letter 0]

8 Sep 2025

PONE-D-25-43674
Safety and efficacy of quick-soluble gelatin microparticles for transarterial embolization of the lower urinary tract in a rabbit urinary bladder embolization model
PLOS ONE

Dear Dr. Chang,

Thank you for submitting your manuscript to PLOS ONE. After careful consideration, we feel that it has merit but does not fully meet PLOS ONE’s publication criteria as it currently stands. Therefore, we invite you to submit a revised version of the manuscript that addresses the points raised during the review process.

The reviewers pointed out specific concerns with the work in particular noting issues with the statistical studies. Please address these in a revision.

We look forward to receiving your revised manuscript.

Kind regards,

Gayle E. Woloschak, PhD

Section Editor

PLOS ONE

Journal Requirements:

**Additional Editor Comments:**

The reviewers noted many specific concerns about the manuscript; the issue about statistics should be corrected either by using a different statistical analysis or justifying why this particular analysis is appropriate over others.

Reviewers' comments:

Reviewer's Responses to Questions

**Comments to the Author**

1. Is the manuscript technically sound, and do the data support the conclusions?

Reviewer #1: Yes

Reviewer #2: Yes

2. Has the statistical analysis been performed appropriately and rigorously? 

Reviewer #1: No

Reviewer #2: Yes

3. Have the authors made all data underlying the findings in their manuscript fully available?

Reviewer #1: Yes

Reviewer #2: Yes

4. Is the manuscript presented in an intelligible fashion and written in standard English?

Reviewer #1: Yes

Reviewer #2: Yes

5. Review Comments to the Author

Reviewer #1: General

Please make sure that all figures and tables are cited in the text in the correct order. Each figure and table should also have a full legend or footnote placed at the end of the document, in an orderly manner. This will make the review process clearer and easier to follow.

The manuscript should be revised to fully follow the ARRIVE 2.0 guidelines for animal research reporting.

Title

Consider adding the word “Preclinical” (or “Pilot Study”) to the title. This will clearly indicate that the study is exploratory and translational in intent, rather than a definitive clinical trial.

Introduction

Please expand on the justification for using rabbits as the experimental model. Provide some background on their urinary tract and vascular anatomy, especially the relevance of the umbilical artery and vesical branches to the embolization procedure.

Clarify whether the main objective of this work is to develop a preclinical model for human applications (for example, future treatment of bladder or urinary tract conditions in people), or whether it is primarily directed toward veterinary applications in animals with bladder or urinary tract disease.

Materials and Methods

The euthanasia schedule should be described more clearly. If 12 rabbits were used, please state explicitly that 3 animals were euthanized at each of the four time-points (0 days, 3 days, 7 days, and 14 days post-embolization). Consider presenting this in a table or study flow diagram to make it easier to understand.

Report the sex distribution of the rabbits. If only one sex was used (for example, only males), explain the reason for this choice. If both sexes were included, indicate the numbers of males and females.

Please provide details of the manufacturer’s composition of the quick-soluble gelatin microparticles (QS-GMP). In addition, provide supporting data or a reference for the dissolution kinetics of the embolic. You state in the Discussion that the particles begin dissolving at about 12 hours and are completely gone by 48 hours. This needs to be supported by either published data, experimental bench testing, or manufacturer specifications.

No a priori sample size or power calculation is presented. Please justify the use of 12 rabbits and 3 animals per group. Either explain what effect size this design was expected to detect or clearly state that the study was exploratory or pilot in nature, and interpret the results with appropriate caution.

Results

Please report the technical success rate of the embolization procedures. Specifically, how many procedures achieved distal embolization in the cranial vesical artery, and how many had to be performed more proximally because the microcatheter could not be advanced far enough.

Describe whether there were any technical problems with catheterization during follow-up procedures.

Indicate whether any medications (such as vasodilators or spasmolytics) were given to manage arterial spasm during embolization.

Clarify exactly how stasis was defined and achieved during embolization. Indicate whether reflux occurred, how it was monitored, and whether it was considered acceptable or a sign of endpoint achievement.

Statistics

The use of the Friedman test is not appropriate in this design. The animals at each time-point are different groups, not repeated measures of the same individuals.

Discussion

In the first paragraph you state that this model “may prove valuable.” Please restate this more clearly in the final paragraph of the Discussion and in the Abstract. Explicitly identify whether the main objective was to establish a preclinical translational model for future human use, or to propose a veterinary clinical application.

Please add explicit reporting of non-target embolization of the ductus deferens. Indicate whether this was observed in your study, whether it was analyzed, and whether it was considered a potential complication.

Reviewer #2: General Comments

This manuscript investigates the safety and efficacy of quick-soluble gelatin microparticles (QS-GMP) in a rabbit urinary bladder embolization model. The study is methodologically well-structured, with comprehensive evaluation using angiography, histopathology, hematological and biochemical analyses. The establishment of a reproducible preclinical bladder embolization model is of clear significance and could contribute to the development of safer embolization strategies for lower urinary tract interventions.　However, there are several issues that need to be addressed before publication. The writing sometimes overstates the novelty or clinical readiness of QS-GMP, whereas the data mainly support feasibility and preliminary safety. Furthermore, while angiographic and histopathological scoring systems are well-described, representative images illustrating these criteria should be included to enhance clarity and reproducibility. Finally, the Discussion could be improved by sharpening the interpretation of findings, avoiding overstatements, and more clearly positioning the contribution of this model within the broader field of embolization research.

Abstract

Line 20: The phrase “promising embolic agents” is overstated at this stage. Since QS-GMP are still in early evaluation, “emerging embolic agents” or “novel embolic agents under investigation” would be more appropriate.

Discussion

Paragraph 1

While the rationale is clear, the paragraph could be strengthened by more explicitly comparing this model to other existing embolization models (e.g., hepatic, renal, prostatic).

Paragraph 5

The discussion of transient hypervascularization is interesting but could be expanded by clarifying whether this represents reactive angiogenesis or an artifact of inflammation. The comparison to hepatic TAE studies is valuable, but it should be noted that urinary bladder physiology may differ substantially.

Paragraph 6

The safety profile is presented convincingly, but the conclusion is slightly overstated. The data support feasibility and short-term safety, but not necessarily long-term safety or clinical application. The language should be moderated accordingly.

Table 1 and Table 4: The scoring system for angiographic findings is clearly described, but representative angiographic images corresponding to each score should be provided. This will help readers understand how the criteria were applied in practice.

Table 2: Similarly, representative histopathological images illustrating each scoring category (edema, necrosis, inflammation, neovascularization, regeneration, etc.) should be included. Without these, it is difficult to validate the scoring method.

6. PLOS authors have the option to publish the peer review history of their article (what does this mean?). If published, this will include your full peer review and any attached files.

Reviewer #1: **Yes: **José Andrés Guirola Ortiz

Reviewer #2: **Yes: **KEN KAGEYAMA

---

## [Author Response · Author response to Decision Letter 1]

12 Oct 2025

Response to Reviewers

Reviewer #1

General

C: Please make sure that all figures and tables are cited in the text in the correct order. Each figure and table should also have a full legend or footnote placed at the end of the document, in an orderly manner. This will make the review process clearer and easier to follow.

R: Thank you for your feedback. We have revised the manuscript to ensure that all figures and tables are cited sequentially in the text. Figure legends and list of tables have been placed at the end of the document.

C: The manuscript should be revised to fully follow the ARRIVE 2.0 guidelines for animal research reporting.

R: We appreciate this important suggestion. We have revised the Materials and Methods section according to the ARRIVE 2.0 checklist.

Title

C: Consider adding the word “Preclinical” (or “Pilot Study”) to the title. This will clearly indicate that the study is exploratory and translational in intent, rather than a definitive clinical trial.

R: As suggested, we have revised the title and abstract to include the term “Preclinical”, indicating the exploratory intent of this work.

Introduction

C: Please expand on the justification for using rabbits as the experimental model. Provide some background on their urinary tract and vascular anatomy, especially the relevance of the umbilical artery and vesical branches to the embolization procedure.

R: As suggested. we have expanded the discussion section to include anatomical relevance of the rabbit umbilical and cranial vesical arteries, add that why this model is appropriate for simulating bladder-selective embolization.

C: Clarify whether the main objective of this work is to develop a preclinical model for human applications (for example, future treatment of bladder or urinary tract conditions in people), or whether it is primarily directed toward veterinary applications in animals with bladder or urinary tract disease.

R: This study was primarily designed as a preclinical pilot investigation intended for veterinary application. To avoid ambiguity, we have revised to clearly state that the current work is veterinary-oriented in intent.

Materials and Methods

C: The euthanasia schedule should be described more clearly. If 12 rabbits were used, please state explicitly that 3 animals were euthanized at each of the four time-points (0 days, 3 days, 7 days, and 14 days post-embolization). Consider presenting this in a table or study flow diagram to make it easier to understand.

R: As your suggestion, we have clarified the euthanasia schedule in the Materials and Methods section. Figure 2 already illustrates the experimental timeline. Should additional visual clarification be required, we would be willing to provide a supplementary version.

C: Report the sex distribution of the rabbits. If only one sex was used (for example, only males), explain the reason for this choice. If both sexes were included, indicate the numbers of males and females.

R: Thank you for highlighting this point. All rabbits used in this study were male. This decision was made because embolization was performed at the level of the umbilical artery rather than the cranial vesical artery. In female rabbits, the uterine artery originates from the umbilical artery, supplying the uterus. Therefore, embolization at this level could have resulted in unintended uterine ischemic injury. To avoid this confounding risk and ensure procedural consistency, only male rabbits were selected. This rationale has been added to the Discussion section for clarification.

C: Please provide details of the manufacturer’s composition of the quick-soluble gelatin microparticles (QS-GMP). In addition, provide supporting data or a reference for the dissolution kinetics of the embolic. You state in the Discussion that the particles begin dissolving at about 12 hours and are completely gone by 48 hours. This needs to be supported by either published data, experimental bench testing, or manufacturer specifications.

R: The QS-GMP used in this study were manufactured from fish-derived gelatin with low molecular weight. The dissolution kinetics are based on the manufacturer’s in-house data as well as previously published in vivo experimental reports. References to support the statement provided in the Discussion section.

C: No a priori sample size or power calculation is presented. Please justify the use of 12 rabbits and 3 animals per group. Either explain what effect size this design was expected to detect or clearly state that the study was exploratory or pilot in nature, and interpret the results with appropriate caution.

R: Thank you for raising this important point. This study was intentionally designed as an exploratory preclinical pilot study aimed at evaluating the feasibility, embolic effect, and tissue response following QS-GMP embolization. Therefore, a formal a priori power calculation was not performed. A total of 12 rabbits (3 per time-point) was selected based on ethical considerations and in accordance with sample sizes commonly used in previous embolization feasibility studies in rabbits. This clarification has been added to the Materials and Methods section.

Results

C: Please report the technical success rate of the embolization procedures. Specifically, how many procedures achieved distal embolization in the cranial vesical artery, and how many had to be performed more proximally because the microcatheter could not be advanced far enough.

R: Thank you for this important clarification. In all 12 procedures, direct advancement of the microcatheter into the cranial vesical artery was not feasible, as our preliminary pilot attempts frequently resulted in catheter wedging or, in some cases, vascular perforation during guidewire manipulation. Therefore, embolization was consistently performed at the level of the umbilical artery in all animals. This level allowed stable catheter positioning without complications. We have clarified this point in both the Results and Discussion sections.

C: Describe whether there were any technical problems with catheterization during follow-up procedures.

R: As your suggestion, we revised technical properties with catheterization during follow-up procedures in angiography section.

C: Indicate whether any medications (such as vasodilators or spasmolytics) were given to manage arterial spasm during embolization.

R: Thank you for raising this point. A diluted vasodilator mixture consisting of 10-fold diluted lidocaine and nitroglycerin (0.1 mL) was infused immediately after microcatheter placement in the umbilical artery in all procedures. This was administered as a precautionary measure rather than in response to observed spasm. No arterial spasm was encountered throughout any procedure. We have added this information to the Materials and Methods section for clarity.

C: Clarify exactly how stasis was defined and achieved during embolization. Indicate whether reflux occurred, how it was monitored, and whether it was considered acceptable or a sign of endpoint achievement.

R: As your suggestion, we have clarified the definition and monitoring of stasis in the Materials and Methods section.

Statistics

C: The use of the Friedman test is not appropriate in this design. The animals at each time-point are different groups, not repeated measures of the same individuals.

R: Thank you for pointing this out. We agree that each time-point comprised independent groups rather than repeated measurements. Accordingly, we have replaced the Friedman test with the Kruskal–Wallis test for overall group comparisons. Our original wording may have unintentionally suggested within-subject comparisons against Day 0; we have revised the text to clarify that Day 0 served only as a reference group, not as a repeated baseline for the same individuals. The Materials and Methods, Results, and Discussion sections have been updated to reflect this change.

Discussion

C: In the first paragraph you state that this model “may prove valuable.” Please restate this more clearly in the final paragraph of the Discussion and in the Abstract. Explicitly identify whether the main objective was to establish a preclinical translational model for future human use, or to propose a veterinary clinical application.

R: Thank you for this valuable comment. As suggested, we have clarified the primary objective of the study. This work was originally designed as a pilot study for veterinary clinical application in lower urinary tract diseases. However, considering the reproducibility and standardized embolization protocol, we agree that the model may also function as a preclinical platform for broader embolization research. We have revised the first and final paragraphs of the Discussion, as well as the concluding sentence of the Abstract and Conclusion.

C: Please add explicit reporting of non-target embolization of the ductus deferens. Indicate whether this was observed in your study, whether it was analyzed, and whether it was considered a potential complication.

R: Although histopathological examination of the ductus deferens and testes was not performed, gross inspection during necropsy revealed no discoloration or structural abnormalities in these organs. We presume that this preservation is due to collateral circulation supplied by the testicular artery branching directly from the abdominal aorta. This clarification has been added to the Discussion as a potential safety consideration regarding non-target embolization.

Reviewer #2: General Comments

C: This manuscript investigates the safety and efficacy of quick-soluble gelatin microparticles (QS-GMP) in a rabbit urinary bladder embolization model. The study is methodologically well-structured, with comprehensive evaluation using angiography, histopathology, hematological and biochemical analyses. The establishment of a reproducible preclinical bladder embolization model is of clear significance and could contribute to the development of safer embolization strategies for lower urinary tract interventions.　However, there are several issues that need to be addressed before publication. The writing sometimes overstates the novelty or clinical readiness of QS-GMP, whereas the data mainly support feasibility and preliminary safety. Furthermore, while angiographic and histopathological scoring systems are well-described, representative images illustrating these criteria should be included to enhance clarity and reproducibility. Finally, the Discussion could be improved by sharpening the interpretation of findings, avoiding overstatements, and more clearly positioning the contribution of this model within the broader field of embolization research.

R: We sincerely appreciate the reviewer’s constructive and encouraging feedback. We have addressed the points you mentioned and made the necessary revisions to our manuscript. According to your suggestions, we have toned down overstatements in the manuscript. In addition, we supplemented the manuscript with representative angiographic and histopathological images to support the scoring systems in the tables.

Abstract

C: Line 20: The phrase “promising embolic agents” is overstated at this stage. Since QS-GMP are still in early evaluation, “emerging embolic agents” or “novel embolic agents under investigation” would be more appropriate.

R: Thank you for pointing this out. We have replaced the phrase “promising embolic agents” with “emerging embolic agents”.

Discussion

Paragraph 1

C: While the rationale is clear, the paragraph could be strengthened by more explicitly comparing this model to other existing embolization models (e.g., hepatic, renal, prostatic).

R: Thank you for the suggestion. We have revised Paragraph 1 of the Discussion to include comparisons with hepatic, renal, and prostatic embolization models.

Paragraph 5

C: The discussion of transient hypervascularization is interesting but could be expanded by clarifying whether this represents reactive angiogenesis or an artifact of inflammation. The comparison to hepatic TAE studies is valuable, but it should be noted that urinary bladder physiology may differ substantially.

R: Thank you for this insightful suggestion. We have revised the paragraph to clarify that the observed hypervascularization may reflect either reactive angiogenesis or inflammatory hyperemia, and we have acknowledged that bladder physiology differs from hepatic circulation, which may influence the vascular response.

Paragraph 6

C: The safety profile is presented convincingly, but the conclusion is slightly overstated. The data support feasibility and short-term safety, but not necessarily long-term safety or clinical application. The language should be moderated accordingly.

R: As noted in our general response, we have toned down the statements in the Abstract, Discussion, and Conclusion to reflect feasibility and short-term safety.

C: Table 1 and Table 4: The scoring system for angiographic findings is clearly described, but representative angiographic images corresponding to each score should be provided. This will help readers understand how the criteria were applied in practice.

Table 2: Similarly, representative histopathological images illustrating each scoring category (edema, necrosis, inflammation, neovascularization, regeneration, etc.) should be included. Without these, it is difficult to validate the scoring method.

R: Thank you for the helpful suggestion. In accordance with the journal’s figure limit, we have added representative angiographic images as Supplementary Figures. These include examples corresponding to each angiographic score category (Tables 1 and 4) and each histopathological scoring parameter (Table 2).

---

## [Decision Letter · Decision Letter 1]

19 Oct 2025

Safety and efficacy of quick-soluble gelatin microparticles for transarterial embolization of the lower urinary tract: preclinical study in a rabbit urinary bladder embolization model

PONE-D-25-43674R1

Dear Dr. Chang:

We’re pleased to inform you that your manuscript has been judged scientifically suitable for publication and will be formally accepted for publication once it meets all outstanding technical requirements.

Kind regards,

Gayle E. Woloschak, PhD

Section Editor

PLOS ONE

Additional Editor Comments (optional):

Thank you for addressing the concerns of the reviewers.

Reviewers' comments:

Reviewer's Responses to Questions

**Comments to the Author**

1. If the authors have adequately addressed your comments raised in a previous round of review and you feel that this manuscript is now acceptable for publication, you may indicate that here to bypass the “Comments to the Author” section, enter your conflict of interest statement in the “Confidential to Editor” section, and submit your "Accept" recommendation.

Reviewer #1: All comments have been addressed

Reviewer #2: All comments have been addressed

2. Is the manuscript technically sound, and do the data support the conclusions?

Reviewer #1: Yes

Reviewer #2: Yes

3. Has the statistical analysis been performed appropriately and rigorously? 

Reviewer #1: I Don't Know

Reviewer #2: Yes

4. Have the authors made all data underlying the findings in their manuscript fully available?

Reviewer #1: Yes

Reviewer #2: Yes

5. Is the manuscript presented in an intelligible fashion and written in standard English?

Reviewer #1: Yes

Reviewer #2: Yes

6. Review Comments to the Author

Reviewer #1: Thank you, this is a clear and well-executed revision. The point-by-point responses and the revised manuscript materially improve clarity and reproducibility.

Reviewer #2: (No Response)

7. PLOS authors have the option to publish the peer review history of their article (what does this mean?). If published, this will include your full peer review and any attached files.

Reviewer #1: **Yes: **JA Guirola

Reviewer #2: **Yes: **Ken Kageyama

---

## [Editor Report · Acceptance letter]

PONE-D-25-43674R1

PLOS ONE

Dear Dr. Chang,

I'm pleased to inform you that your manuscript has been deemed suitable for publication in PLOS ONE. Congratulations! Your manuscript is now being handed over to our production team.

Kind regards,

on behalf of

Dr. Gayle E. Woloschak

Section Editor

PLOS ONE